# Improved 3D cellular morphometry of *Caenorhabditis elegans* embryos using a refractive index matching medium

**Rain Xiong**[1,2], **Kenji Sugioka**[1,2]*

**1** Life Sciences Institute, The University of British Columbia, Vancouver, BC, Canada, **2** Department of Zoology, The University of British Columbia, Vancouver, BC, Canada

* sugioka@zoology.ubc.ca

**Data Availability Statement:** All relevant data are within the manuscript, Supporting Information files, and via Figshare: https://doi.org/10.6084/m9.figshare.c.5108522.v1.

## Abstract

Cell shape change is one of the driving forces of animal morphogenesis, and the model organism *Caenorhabditis elegans* has played a significant role in analyzing the underlying mechanisms involved. The analysis of cell shape change requires quantification of cellular shape descriptors, a method known as cellular morphometry. However, standard *C. elegans* live imaging methods limit the capability of cellular morphometry in 3D, as spherical aberrations generated by samples and the surrounding medium misalign optical paths. Here, we report a 3D live imaging method for *C. elegans* embryos that minimized spherical aberrations caused by refractive index (RI) mismatch. We determined the composition of a refractive index matching medium (RIMM) for *C. elegans* live imaging. The 3D live imaging with the RIMM resulted in a higher signal intensity in the deeper cell layers. We also found that the obtained images improved the 3D cell segmentation quality. Furthermore, our 3D cellular morphometry and 2D cell shape simulation indicated that the germ cell precursor $P_4$ had exceptionally high cortical tension. Our results demonstrate that the RIMM is a cost-effective solution to improve the 3D cellular morphometry of *C. elegans*. The application of this method should facilitate understanding of *C. elegans* morphogenesis from the perspective of cell shape changes.

## Introduction

Live imaging of *Caenorhabditis elegans* (*C. elegans*) embryos plays a paramount role in dissecting questions pertaining to both cell and developmental biology. However, widely used *C. elegans* live imaging methods with conventional confocal microscopes fails to capture entire embryos and embryonic cell sets at high resolution presumably due to spherical aberrations. Spherical aberrations misalign the optical paths, resulting in the loss of signal intensity and resolution. This effect increases with sample thickness, limiting the capacity of 3D cellular morphometry [1]. A major source of spherical aberration is the refractive index (RI) mismatch among three different materials: the lens immersion medium, samples, and the medium surrounding the samples. In typical *C. elegans* live imaging, samples are mounted onto 2%-4% agarose and are bathed in M9 or egg salt buffer [2, 3], or bathed in these buffers without

**Funding:** This work was supported by the Canadian Institutes of Health Research (Project Grant; PJT-169145) to K.S. *C. elegans* strains were provided by the *Caenorhabditis* Genetics Center (funded by the NIH Office of Research Infrastructure Programs; P40 OD010440).

**Competing interests:** The authors have declared that no competing interests exist.

agarose [4]. These commonly used aqueous media (RI: ~1.33) should create the RI mismatch with both lens immersion media (RI: 1.52 for oil immersion) and *C. elegans* embryos (RI: 1.33–1.37; [5]).

In this study, we focused on the cost-effective method to reduce the RI mismatch that can be immediately adopted by the broad *C. elegans* research community. It is known that the use of water (RI: 1.33) and silicone (RI: 1.4) as lens-immersion media reduces the RI mismatch, but high-numerical aperture (NA) objective lenses designed for water- and silicone-immersion typically cost more than 10,000 USD. The use of non-conventional microscopy systems such as dual-view plane illumination microscopy [6] and lattice light-sheet microscopy [7] allow rapid and high-resolution volumetric imaging. However, these are also currently not cost-effective solutions.

We hypothesized that a refractive index matching medium (RIMM), which has a RI close to that of the specimen, reduces spherical aberration during *C. elegans* 3D live imaging with conventional confocal microscopy. Recently, Boothe et al. identified iodixanol solution as a RIMM suitable for live imaging of cultured cells, zebrafish embryos, and planarians [8]. Iodixanol is a transparent solution with a high RI of 1.429 at a 60% stock solution. Booth et al. diluted 60% iodixanol solution to identify the RIMM for each sample. However, the RIMM for *C. elegans* has not been determined.

In this study, we determined the composition of a RIMM for *C. elegans* live imaging. To analyze the RI mismatch between *C. elegans* samples and the surrounding medium, we imaged embryos and adult heads in different iodixanol concentrations using Nomarski differential interference contrast microscopy (DIC), a common transmitted light microscopy method used by worm researchers. Using the obtained RIMM, we performed 3D volumetric imaging and 3D watershed segmentation to generate cellular surface models. These models were then used to perform 3D cellular morphometry of individual embryonic blastomeres.

## Materials and methods

### *C. elegans* culture and strains

All strains used in this study were cultured using standard methods [9]. All worms were grown at 25˚C and imaged at 22.5˚C. The following integrated transgenic lines were used: *cpIs56* for TagRFP::PH [10] and *ruIs32* [11] for GFP::H2B.

### Microscopy

All embryos were dissected in an egg salt buffer from gravid hermaphrodites [12]. For live imaging, embryos were transferred by a mouth pipette into the egg salt buffer containing different iodixanol concentrations [13]. We obtained similar results by directly dissecting gravid hermaphrodites in the iodixanol solution. Owing to the high density of iodixanol solutions, embryos tend to float in the medium. To mitigate this problem and situate embryos as close as the coverslips as possible, we used polystyrene beads with embryo thickness as spacers between coverslips and glass slides (30 μm diameter, Kisker Biotech GmbH & Co). *C. elegans* adult samples were also prepared with the beads. Nomarski DIC imaging was performed with generic CCD cameras mounted on either AxioPlan or AxioSkop compound microscopes (Zeiss). Confocal images used to generate a right graph in Fig 2D and a graph in Fig 2E were captured with a spinning-disk confocal unit CSU-W1 (Yokogawa) and a scientific CMOS camera Prime 95B (Photometrics) mounted on an inverted microscope Olympus IX83 (Olympus), controlled by Cellsense Dimension (Olympus). Samples were illuminated by a diode-pumped laser with 561 nm wavelength. Z-stacks were obtained with a silicone-immersion UPLSAPO60XS2 objective lens (60X, NA = 1.3; Olympus) at 0.5-μm Z-step and with a silicone-immersion oil (RI: 1.41 at

23˚C; Z81114, Olympus). The rest of confocal images were captured with a spinning-disk confocal unit CSU-W1 with Borealis (Andor Technology) and dual EMCCD camera iXon Ultra 897 (Andor Technology) mounted on an inverted microscope Leica DMi8 (Leica Microsystems), controlled by Metamorph (Molecular Devices). Samples were illuminated by diode-pumped lasers with 488 nm and 561 nm wavelengths. Z-stacks were obtained with an oil-immersion PL APO objective lens (63X, NA = 1.4; Leica) at 0.5-μm Z-step and with an immersion oil (type F, RI = 1.51; Leica).

## Determination of a RIMM for live *C. elegans*

A 60% iodixanol solution (Optiprep; D1556, Millipore Sigma) was sequentially diluted with egg salt buffer. Embryos at mixed stages and adult head regions in different iodixanol concentrations were then imaged with a 10X Plan-Apochromat objective lens (NA = 0.45; Zeiss). Using Fiji [14], we drew the regions of interest (ROI) corresponding to the whole embryos and adult heads and quantified the signal intensities. The maximum gray values were divided by the minimum gray values to calculate the image contrast. The RI of commonly used *C. elegans* buffers were measured at 22.5˚C using a digital refractometer (13940000, Reichert) at a wavelength of 598 nm. Egg salt buffer, M9 buffer, and Shelton's growth medium were prepared as described before [12, 15].

## Eggshell penetration assay

To test whether molecules with the size of iodixanol can pass the eggshell, we mounted wild-type embryos in the 30% iodixanol containing 0.05 U/μl Alexa Fluor 568 Phalloidin (1590 Da; A12380, Invitrogen) and polystyrene beads. Immediately after the preparation of the sample slide as described above, confocal and brightfield images were captured.

## Quantification of the signal intensity

For the generation of the binary images shown in Fig 2C, we used Otsu's thresholding method with Fiji [16]. To calculate the signal intensity of TagRFP-PH in each image, we drew a 4.96 μm line passing the cell-cell boundary and generated a plot profile of gray value (S4A Fig). Mean signal intensity of the first 1 μm length, corresponding to the cytoplasmic region, was used as a background value. Signal intensity was then calculated by the following equation.

$$[Signal\ intensity] = \frac{[gray\ value]}{[background\ value]} - 1$$

Each dot in Fig 2E represents the maximum signal intensity within a cell-cell boundary. Note that four to five cell-cell boundaries were measured for each embryo.

## 3D segmentation and generation of cellular surface models

To perform 3D segmentation of embryos, source images (TagRFP::PH; 0.16 μm/pixel and 0.5-μm voxel depth) were processed using Fiji. First, images were processed with the Attenuation Correction plug-in ([17]; opening radius = 3.0, reference slice was set to midplane) to normalize the signal intensity across different depths. Second, noise was attenuated with Gaussian blur 3D to facilitate the detection of cell-cell boundaries (X sigma = 2, Y sigma = 2, and Z sigma = 0.6). Third, images were processed using the morphological segmentation function of the MorphoLibJ plugin ([18]; image type is "border," tolerance = 22). Obtained catchment basin images (right panels in Fig 3B and 3C) were then analyzed for cellular sphericity using the "analyze regions" function of the MorphoLibJ plugin. Finally, catchment basin images

were used to generate 3D surface models of the embryos using an ImageJ 3D viewer macro [18]. To generate images shown in Figs 4A, 4C–4E and 3D surface models were exported as STL files and rendered using Adobe Photoshop (Adobe).

## 2D cell shape simulation

To perform 2D cell shape simulation, we have used the previously reported model [19] and a simulation tool called Morphogenie (http://www.celldynamics.org/celldynamics/downloads/morphogenie/index.html). First, we defined the position and size of thirteen cells by specifying the vertices and junctions in a two-dimensional space. Next, posteriorly located three cells were specified as "E.p", "P$_4$", and "D" in a pattern resembles that of wild-type embryos. Other cells were specified as the same cell type ("Others"). Cells' cortical tension ("outer boundary tension") was set to 0.1 for all the cell types except for the P$_4$ cell. Tension at the cell-cell boundary ("inner boundary tension") was set to 0.1 for all cell-boundaries. In order to test the effects of P$_4$ cortical tension on the P$_4$ internalization, we used the P$_4$ cortical tension 0.1, 0.15, and 0.2.

## Statistical analysis

For multiple comparisons, a one-way ANOVA with the Holm-Sidak's method was used. Other analyses were performed using the Welch's t-test. Statistical tests were performed using Prism 8 (Graph Pad). No statistical method was used to predetermine the sample size. The experiments were not randomized. The investigators were not blinded to the study.

## Results

### 30% iodixanol is a RIMM for live *C. elegans* embryos

To determine the composition of a RIMM for *C. elegans* embryos, we imaged embryos bathed in egg salt buffer containing 0%, 10%, 20%, 25%, 30%, 40%, 50%, and 60% iodixanol (Fig 1 and S1 Fig). A previous study used phase-contrast microscopy to determine the optimal iodixanol concentration, because it converts the local differences in refractive indices of transparent specimens into the image contrast [8]. The loss of image contrast indicates the RI matching between samples and the surrounding medium. In this study, we used Nomarski DIC microscopy as it is widely used by *C. elegans* researchers. DIC microscopy converts the local differences in thickness and refractive indices of transparent specimens into the image contrast [20]. *C. elegans* embryos are similar in thickness (30 μm; [21]), such that the changes in DIC image contrast should reflect the degree of RI mismatch between embryos and the surrounding medium. Thus, we sought a condition where the image contrast, calculated by the ratio of maximum to minimum signal intensity, were at their lowest (Fig 1A and 1C). As a reference for the DIC shear direction, and as spacers against compression, we added 30 μm polystyrene beads to the medium (spheres in Fig 1A). We found that embryos imaged at 20%, 25%, and 30% iodixanol showed the lowest image contrast (Fig 1A and 1C). The 30% iodixanol solution did not affect embryo viability (0% iodixanol: 98% hatched, n = 154; 30% iodixanol: 97% hatched, n = 148).

We assessed whether iodixanol could pass the *C. elegans* eggshell. *C. elegans* embryo is surrounded by the chitinous eggshell and the permeability barrier [22]. To reduce the RI mismatch between embryos and the surrounding media, 30% iodixanol may need to penetrate into the perivitelline space, a space between the eggshell and the permeability barrier (S2 Fig; a space between a white solid line and a yellow dotted line). A previous study reported that 389 Da and 1300 Da fluorescent dye passed the eggshell while that of 3000 Da did not [22]. The molecular weight of iodixanol is 1550 Da, which is only slightly higher than the reported limit of 1300 Da, such that iodixanol likely passes the eggshell. To directly test if molecules with the

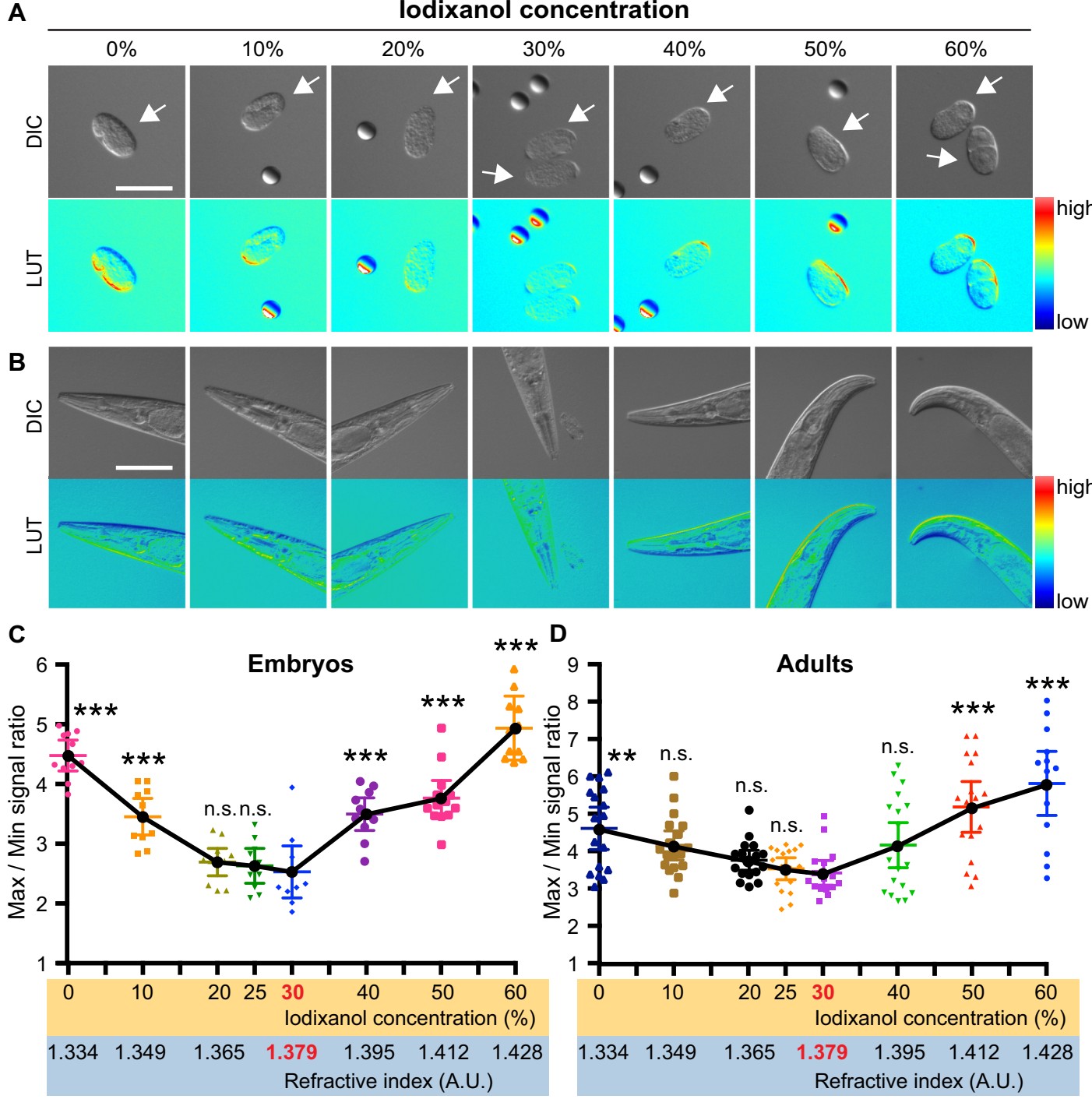

**Fig 1. Determination of a refractive index matching medium for live *C. elegans*.** (A, B) *C. elegans* embryos (A) and adult heads (B) imaged at different iodixanol concentrations. Images were obtained by Nomarski DIC microscopy. LUT images are intensity-based pseudocolored DIC images (pseudocolor was applied using the look-up table called "physics"). Arrows indicate embryos. Note that the spheres in some images are polystyrene beads used as a reference (see Materials and methods). (C, D) The extent of image contrast was calculated by dividing the maximum signal intensity in the sample area by that of the minimum. As both curves show that the average image contrast is the lowest at 30% iodixanol, statistical tests were performed by comparing samples treated with 30% iodixanol and others. P-values were calculated by one-way ANOVA with the Holm-Sidak's multiple comparison test. n.s. ($P > 0.05$), ** ($P < 0.001$), and *** ($P < 0.0001$). Scale bars, 50 μm.

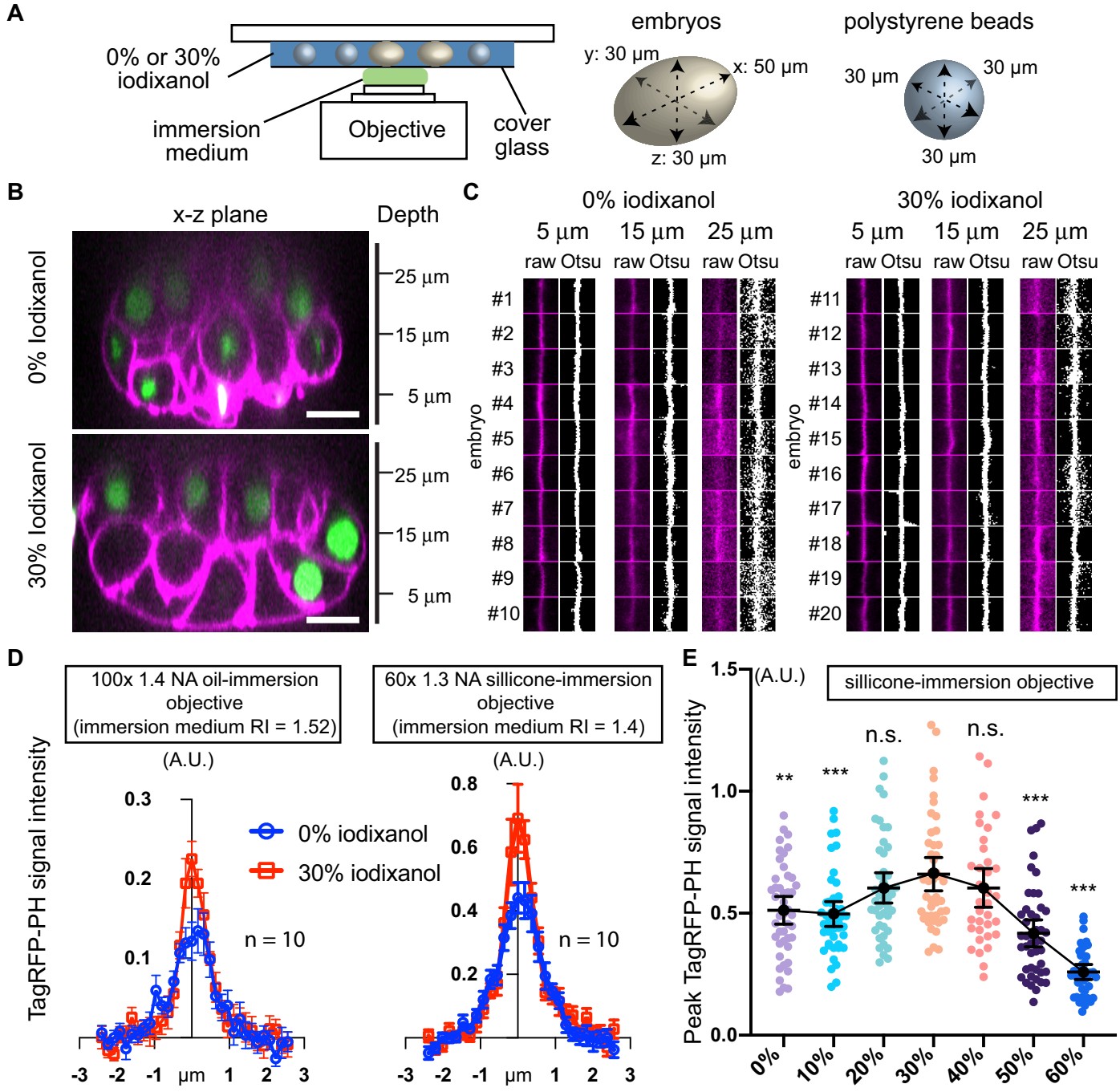

**Fig 2. The refractive index matching medium improves the signal intensity in the deeper cell layers.** (A) Schematic drawings of the imaging setup. Embryos and embryo-sized polystyrene beads were sandwiched between coverslips and glass slides to prevent embryo flotation due to the higher density of iodixanol solution. (B, C) Depth-induced blurring of the cell-cell boundary. Spinning-disk confocal images of 24-cell stage embryos (B). Magenta, TagRFP-PH (plasma membrane); green, histone H2B-GFP (chromosomes). Plasma membrane signal at the cell-cell boundary (C). Each raw image in C, which captures a cell-cell boundary in 4.96 µm x 4.96 µm area, was converted into binary images using Otsu's method. (D) Intensity profiles of cell boundaries. A line passing a cell-boundary at 25-µm depth was used to measure intensity profiles of TagRFP-PH. The background level was adjusted to zero (see S4A Fig for detail). Error bars are standard errors of means. (E) Peak Tag RFP-PH signal intensity at 25-µm depth in different iodixanol concentrations. Error bars are 95% confidence intervals. Statistical test was performed by comparing samples treated with 30% iodixanol and others. P-values were calculated by one-way ANOVA with the Holm-Sidak's multiple comparison test. n.s. (P > 0.05), ** (P < 0.001), and *** (P < 0.0001). All depths shown are relative to the cell surface closest to the objective lens. Scale bars, 10 µm.

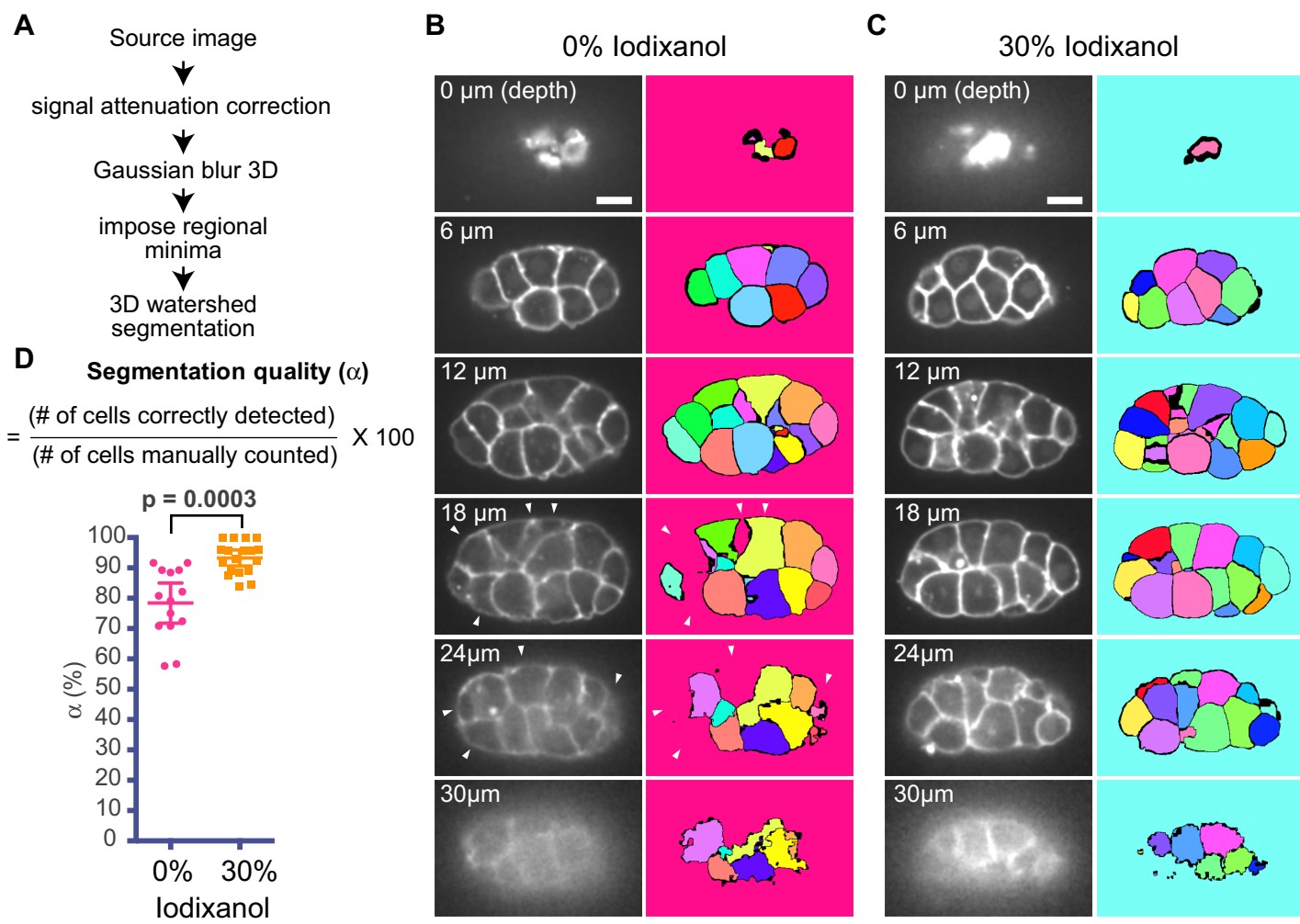

**Fig 3. A refractive index matching medium improves 3D segmentation quality.** (A) Image processing pipeline used in this study. See text. (B, C) 3D watershed segmentation of *C. elegans* 24-cell stage embryos using 0% (B) and 30% (C) iodixanol. The left and right panels show the images after the 3D Gaussian blur and 3D watershed segmentation in the pipeline, respectively. Note that each segmented cell was labeled with a unique color. Arrowheads indicate cells with segmentation errors. (D) Quantified segmentation quality: by using the number of cells manually counted as a ground truth, segmentation quality α was determined. P-values were calculated using the Welch's t-test. Scale bars, 10 μm.

size of iodixanol can pass the eggshell, we bathed embryos into 30% iodixanol containing Alexa 568-phalloidin (1590 Da). We found that Alexa 568-phalloidin passed the eggshell (S2 Fig). Although the fluorescent molecules did not pass the permeability barrier, most areas of the embryo were in very close proximity to the surrounding perivitelline space (S2 Fig).

We also determined the composition of a refractive-index-matching medium for *C. elegans* adults. We used the head region of adult *C. elegans* to minimize variations in sample thickness. Similar to embryos, using 30% iodixanol resulted in the lowest image contrast (Fig 1B and 1D). L4-stage larva cultured in 30% iodixanol solution with OP50 E. coli produced fertile progenies, suggesting that the RIMM was not particularly detrimental to *C. elegans* postembryonic development. The RI of the 30% iodixanol solution was 1.379 ± 0.00004 at 22.5˚C according to the measurement using a refractometer at 589 nm wavelength (S3 Fig). Selection of buffers should not have a large impact, since the RI of commonly used *C. elegans* buffers, such as M9 (RI: 1.335 ± 0.0001), egg salt (RI: 1.334 ± 0.00009), and Shelton's growth medium (RI:

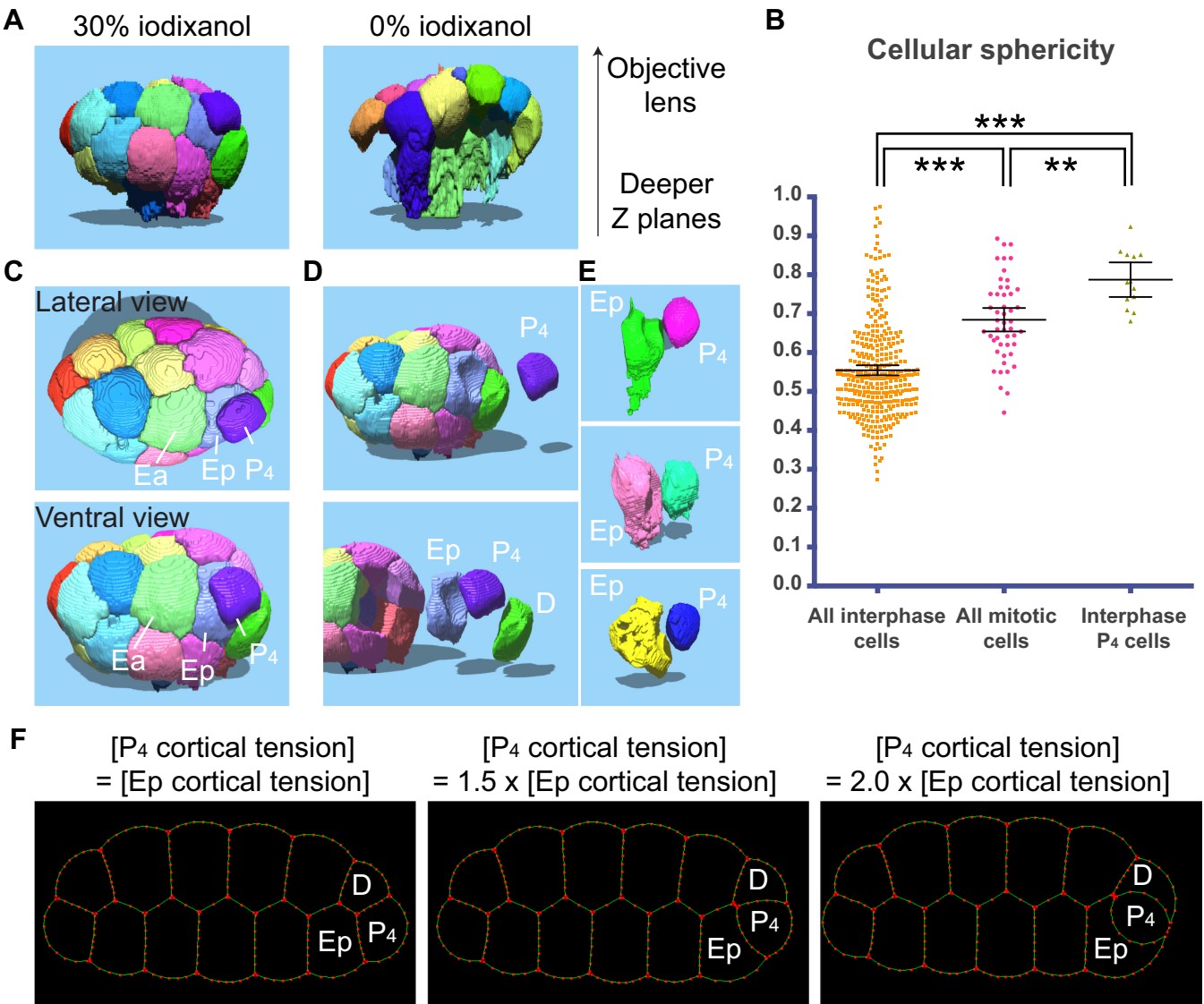

**Fig 4. Cellular morphometry analysis predicts exceptionally high cortical tension in the interphase P$_4$ blastomere.** (A) Cell surface models reconstructed using 3D segmentation data. Note that in 0% iodixanol, segmentation failed, and cells elongated in the deeper cell layers. (B) Cellular sphericity of 413 blastomeres from 18 embryos is shown. Error bars indicate 95% confidence interval (CI). P-values were calculated by one-way ANOVA with the Holm-Sidak's multiple comparison test. ** (P < 0.001) and *** (P < 0.0001). (C) Lateral and ventral views of surface model of 24-cell stage embryos obtained with 30% iodixanol. (D, E) Surface model of P$_4$ and E.p cells from four different embryos. (F) 2D simulation of cellular shape changes. Each cell-cell boundary contains more than two contractile and viscoelastic elements to mimic active force generation and cell deformability of the cell cortex (see text). Cortical tension in P$_4$ cells gradually increased relative to that in E.p and D cells.

1.339 ± 0.00005) were similar at 22.5˚C (S3 Fig). Based on these results, we concluded that 30% iodixanol, diluted in any common *C. elegans* buffers, is a RIMM for both *C. elegans* embryos and adult head regions.

## The RIMM improves the signal-to-noise ratio in the deeper cell layers

We tested whether the obtained RIMM improved 3D live imaging of *C. elegans* embryos. One of the challenges in volumetric imaging is the reduced signal-to-noise ratio in deeper cell layers distal to the objective lens. This is mainly caused by the depth- and RI mismatch-induced

spherical aberration that leads to the loss of contrast and signal intensity [1]. A previous study reported that an iodixanol-based RIMM improved the signal intensity in the deeper cell layers in other organisms by reducing the RI mismatch between samples and the surrounding media [8]. The same study used a silicone-immersion objective lens to minimize the RI mismatch between 1) the lens-immersion medium and samples and 2) the lens-immersion medium and the medium surrounding samples. However, a silicone-immersion objective lens is not a cost-effective measure for a wide-research community. We hypothesized that a RIMM alone could improve spherical aberrations regardless of the types of objective lenses by contributing to the reduction of the RI mismatches among the lens-immersion medium, samples, and the medium surrounding samples. Thus, we performed live imaging of histone H2B::GFP (chromosome) and TagRFP::PH (plasma membrane) expressed in *C. elegans* embryos with a spinning-disk confocal microscope equipped with an oil-immersion objective lens. Since embryos float in the 30% iodixanol solution, the distance between the objective lens and the embryos can differ when using 0% and 30% iodixanol. To minimize these differences in spherical aberration, we used polystyrene beads with embryo thickness as spacers (Fig 2A). In this imaging condition, the longitudinal surface of embryos physically contacts with coverslips, and all embryos are situated at a similar depth regardless of iodixanol concentration (Fig 2A and 2B).

We assessed the signal intensity of TagRFP-PH at different depths and different iodixanol concentrations. We collected multiple 4.96 μm x 4.96 μm images at 5-μm, 15-μm, and 25-μm depth, with each capturing a cell-cell boundary in the center (Fig 2C). First, we qualitatively assessed the outcomes of automatic image thresholding using Otsu's method [16]. At 5-μm and 15-μm depth, cell-cell boundaries were clearly defined in both 0% and 30% iodixanol. On the other hand, cell-cell boundaries at 25-μm retained better defined linear patterns in 30% iodixanol (Fig 2C). Second, we quantitatively analyzed the TagRFP-PH signal intensity. We drew a line that perpendicularly passes the cell-cell boundary and generated intensity profiles (Fig 2D and S4A Fig). The signal intensities at 5-μm and 15-μm were not improved by 30% iodixanol (S4B Fig). At 25-μm depth, the cell-cell boundary showed a higher signal intensity with 30% iodixanol (Fig 2D; left graph). The average peak signal intensity with 30% iodixanol was 0.2566 ± 0.07, which was 1.38-fold higher than that with 0% iodixanol (0.1854 ± 0.070, p = 0.0388 by Welch's t-test, n = 10 for each condition). These results suggest that the RIMM alone improves the signal intensity in the deeper cell layer in *C. elegans* embryos.

We also assessed whether the observed improvement could be achieved with a silicone-immersion objective lens. Consistent with the result using an oil-immersion objective lens and the previous study by Boothe et al. [8], we found that the RIMM improved the signal intensity at 25-μm depth (Fig 2D; right graph). In order to confirm the optimal iodixanol concentration for *C. elegans* fluorescence imaging, we have compared the peak TagRFP-PH signal intensities at 25-μm using different iodixanol concentrations (Fig 2E). The average peak signal intensity was the highest at 30% iodixanol concentration, confirming that the DIC-based RI matching method was successful in determining the RIMM (Fig 1C and Fig 2E). Taken together, these results suggest that a RIMM can improve signal intensity at the deep cell layer regardless of the types of objective lens immersion media.

## The RIMM improves the 3D segmentation quality

Next, we tested whether the RIMM improved the 3D segmentation quality of individual cells in embryos. We imaged tagRFP-PH in 24- to 32-cell stage embryos for whole embryo volume (30-μm depth, Z step = 0.5 μm) and performed 3D segmentation of cells using the membrane signals as source images. We processed the source images using algorithms such as signal attenuation correction [17], Gaussian blur, and imposing regional minima [18]

(Fig 3A; see Materials and methods). The images after Gaussian blur are shown in the left panels in Fig 3B and 3C for embryos bathed in 0% and 30% iodixanol media, respectively. Processed images were then segmented using a 3D watershed segmentation algorithm [18] (Fig 3A). When the cells were successfully segmented, individual blastomeres were labeled by unique colors, as shown in Fig 3B and 3C (right panels). From the surface plane to the mid-plane, segmentation quality appeared to be similar between 0% and 30% iodixanol media (Fig 3B and 3C; right panels). However, segmentation errors occurred more frequently in the deeper cell layers when using 0% iodixanol (Fig 3B; white arrowheads). To quantify the 3D segmentation quality, we manually counted the number of cells in embryos and used these counts as the ground truth. The manually counted cell number was then divided by the cell number detected by 3D segmentation to calculate the segmentation quality denoted as α (Fig 3D). We found that α obtained for 30% iodixanol was 91% (n = 19), while that for 0% iodixanol was 78% (n = 14; Fig 3D). These results suggest that the RIMM improves 3D segmentation quality.

## Improved 3D segmentation allows high-throughput cellular morphometry analysis

We sought to determine the method by which improved 3D segmentation could facilitate the analysis of developmental processes. We generated 3D surface models of individual blastomeres in embryos using the results of 3D watershed segmentation (Fig 4A; see Materials and methods). In addition to the segmentation errors described above, we found that segmented cells in the deeper cell layers were more abnormally elongated in 0% iodixanol, likely due to the spherical aberration (Fig 4A). Thus, the RIMM allowed us to obtain more accurate and comprehensive cellular morphology data.

Next, we examined how meaningful information could be extracted by using cellular morphometry. As a proof of principle, we analyzed cell sphericity in 20- to 32-cell stage embryos. Owing to the improved 3D segmentation, we were able to analyze the sphericity of 413 individual blastomeres from 18 embryos by 3D ellipsoid fitting [18] (Fig 4B). As expected, on average, mitotic cells had higher sphericity than interphase cells due to mitotic rounding (Fig 4B; [23]). However, we found that interphase $P_4$ blastomeres exhibited higher sphericity than mitotic cells (Fig 4B). We confirmed this data by manually inspecting the 3D surface models of blastomeres (Fig 4C–4E). The $P_4$ blastomere is a germline cell surrounded by its sister cell, known as D, and an endoderm precursor cell E.p (Fig 4C and 4D). We confirmed that $P_4$ cells were round and bulged into E.p and D cells, forming curved cell-cell boundaries (Fig 4E). As reported previously, curved cell-cell boundaries are induced by the asymmetry of cortical tension and internal cellular pressure [24]. In *C. elegans* 2-cell stage embryos, the anterior AB cell bulges into the posterior $P_1$ cell due to the higher cortical tension in AB [24]. Thus, we speculated that interphase $P_4$ cells had higher cortical tension than E.p and D cells. To test this possibility, we employed a 2D simulation of cell shape change using a previously described model [19]. In this simulation, the dynamics of cell shape change were predicted based on the tension values at different cell-cell boundaries. Each cell-cell boundary contains more than two smaller contractile and viscoelastic elements. These elements, with fixed tension and viscosity, mimic actomyosin contraction and resistance to deformation, respectively. Under this condition, we gradually increased the cortical tension in the $P_4$ cell relative to those in E.p and D cells (Fig 4F). We found that $P_4$-E.p and $P_4$-D boundaries started to curve when the $P_4$ cortical tension was 1.5 fold higher than in the other cells, and that $P_4$ cells bulged into E.p and D cells (Fig 4F). These results, obtained by 3D cellular morphometry, suggest that the interphase $P_4$ cell retains unexpectedly high cortical tension.

## Discussion

In this study, we determined the composition of an RIMM for *C. elegans* embryos and adult heads. The obtained RIMM improved the 3D segmentation quality and enabled more accurate and high throughput 3D cellular morphometry analysis. Based on the cellular morphometry analysis performed, we identified that interphase $P_4$ cells have exceptionally higher cortical tension than the surrounding cells.

From our experiments, the RI suitable for *C. elegans* live imaging medium is estimated to be 1.379 for embryos and adult heads. In the RI matching method using DIC, only variables were concentrations of iodixanol, such that the method is sensitive to the RI mismatch between samples and the surrounding media, but not to the internal heterogeneity of RI within specimens. The RI value for adult heads, which was determined by the previous tomographic phase microscopy analysis (RI: 1.36–1.38; [25]) and the 3D refractive index microscopy analysis (RI: ~1.35; [5]), were slightly different compared to the value obtained by us (RI: 1.379). Similarly, the RI value for embryos estimated by 3D refractive index microscopy was highly heterogeneous (RI: 1.33–1.37; [5]). However, the reported RI values at the outer edge of adults (all regions, including the head) and embryos were around 1.37, which was similar to the value obtained in our study. We found that iodixanol did not pass the permeability barrier (S2 Fig), suggesting that the physical interaction between embryo and the RIMM is unnecessary for reducing spherical aberrations. As one of the primary sources of depth-dependent spherical aberrations originates from the RI mismatch between the samples and the surrounding media, we conclude that the RI matching between the outer edge of embryos and the RIMM in the perivitelline space was sufficient to improve the spherical aberrations.

The use of a RIMM in conjunction with a water- or a silicone- immersion objective lens would generate the best 3D segmentation results for confocal microscopy. This is because these objective lenses minimize the refractive index mismatch between 1) the objective lens immersion medium and samples and 2) the objective lens immersion medium and the medium surrounding samples. We have not compared the performance between water-immersion and silicone-immersion objectives, but the silicone-immersion objective would be better because of its higher numerical aperture and the very low evaporation rate of silicone. Alternatively, the use of dual-view plane illumination microscopy or lattice light-sheet microscopy would allow generation of isotropic images. However, all measures discussed are costly and is not always accessible to many researchers and institutes. Our study, using conventional spinning-disk confocal microscopy and an oil-immersion lens, demonstrated that a RIMM alone was highly effective at improving 3D segmentation quality. With this low-cost solution, university instructors can also print 3D models and use them for biology curricula. It should be noted that a RIMM improves signal intensity at the image acquisition step, such that it can be coupled with other state-of-the-art post-acquisition 3D cell segmentation algorithms, such as RACE [26], 3DMMS [27], and BCOMS [28].

Cellular morphometry analysis revealed that the interphase $P_4$ cell had an exceptionally high sphericity and bulges into the E.p cell. As confirmed in the 2D cell shape simulation, the high sphericity of $P_4$ is probably due to the high cortical tension reminiscent of mitotic rounding. As mitotic rounding accelerates cell invagination [29], $P_4$ rounding may accelerate its internalization. Indeed, the daughters of $P_4$, primordium germ cells (PGCs), migrate into the interior of embryos by hitchhiking gastrulation [30]. Thus, the cortical tension of $P_4$ and its descendants may be precisely controlled to promote the PGC internalization process. Further analysis is required to test these possibilities, but our study demonstrates that a RIMM facilitates 3D cellular morphometry analysis in *C. elegans*.

## Supporting information

**S1 Fig.**
(TIF)

**S2 Fig.**
(TIF)

**S3 Fig.**
(EPS)

**S4 Fig.**
(EPS)

## Acknowledgments

We thank Bruce Bowerman for providing valuable advice. We thank Chris Doe for sharing lab equipment. We thank the *Caenorhabditis* Genetics Center (funded by the NIH Office of Research Infrastructure Programs; P40 OD010440) for *C. elegans* strains. We thank Farid Jalali for critical reading of the manuscript. This work was supported by the Canadian Institutes of Health Research (Project Grant; PJT-169145) to K.S.

## Author Contributions

**Conceptualization:** Kenji Sugioka.

**Data curation:** Kenji Sugioka.

**Formal analysis:** Rain Xiong, Kenji Sugioka.

**Funding acquisition:** Kenji Sugioka.

**Supervision:** Kenji Sugioka.

**Writing – original draft:** Rain Xiong, Kenji Sugioka.

**Writing – review & editing:** Kenji Sugioka.

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
