## [Decision Letter · Decision Letter 0]

9 Jul 2020

PONE-D-20-18786

Improved 3D cellular morphometry of Caenorhabditis elegans embryos using a refractive index matching medium

PLOS ONE

Dear Dr. Sugioka,

Thank you for submitting your manuscript to PLOS ONE. After careful consideration, we feel that it has merit but does not fully meet PLOS ONE’s publication criteria as it currently stands. Therefore, we invite you to submit a revised version of the manuscript that addresses the points raised during the review process.

You will see that Reviewer 1 raised a substantive concern about whether the correction of spherical aberration in deep sections can occur if the iodixanol is not penetrating the eggshell, and noted that the degree of improvement appears small. We would like to give you a chance to address this concern and other issues raised by the reviewers in your manuscript. Please also note that the statistical test used in Figure 2D, the first figure showing the improvement quantitatively, is a test that assumes a normal distribution of the data.

We look forward to receiving your revised manuscript.

Kind regards,

Bob Goldstein

Academic Editor

PLOS ONE

Journal Requirements:

Reviewers' comments:

Reviewer's Responses to Questions

**Comments to the Author**

1. Is the manuscript technically sound, and do the data support the conclusions?

Reviewer #1: No

Reviewer #2: Yes

Reviewer #3: Yes

2. Has the statistical analysis been performed appropriately and rigorously? 

Reviewer #1: N/A

Reviewer #2: Yes

Reviewer #3: Yes

3. Have the authors made all data underlying the findings in their manuscript fully available?

Reviewer #1: Yes

Reviewer #2: No

Reviewer #3: Yes

4. Is the manuscript presented in an intelligible fashion and written in standard English?

Reviewer #1: Yes

Reviewer #2: Yes

Reviewer #3: Yes

5. Review Comments to the Author

Reviewer #1: The manuscript by Xiong & Sugioka describes the use of a refractive index matching medium to increase the visibility of in vivo fluorescence images of C. elegans embryos in order to facilitate the analysis of cellular morphometry. Although there are some potentially interesting aspects to this study, I consider it fundamentally flawed and not worthy of publication in its current form.

The eggshell of a C. elegans embryo is very impermeable and, as such, it is very unlikely that any of the iodixanol refractive index matching agent that was used will penetrate the embryo. This means that the only effect that refractive index matching the external medium will have is to reduce scattered light from the bathing medium/eggshell boundary. The claimed correction of spherical aberration in deep sections cannot occur. In Fig. 2 the authors compare two embryos with different refractive index bathing media. There is very little difference to be seen, and both images exhibit the tell-tale deep section axial expansion typical of spherical aberration. There is also little visible improvement of deep section images to be seen in Fig. 3. Given the intrinsic variability of in vivo fluorescence between embryos and the fact that different embryos were used in the comparisons, I am unconvinced by these data. There is a well-established way of reducing the spherical aberration from in vivo specimens that the authors do not mention, and that is to use water immersion lenses, where there is a reduced refractive index mismatch between cytosol and the lens immersion medium.

I did find the author’s use of DIC microscopy to tune the refractive index of the medium interesting. As they demonstrate, this reduces the contrast extremes at the boundary of the embryo while maintaining the DIC contrast of structures within the embryo. This strategy might improve visibility of cells adjacent to the lateral sides of the embryo and thereby facilitate cell lineage studies. This simple technique might be worth developing into a technical note.

Reviewer #2: Summary

The authors of this study apply refractive index media matching (RIMM), previously reported in Boothe, et al. 2017 for other organisms, to imaging of C. elegans embryos. They empirically and quantitatively determined the RIMM by titrating different percentages of iodixanol into egg salts, a commonly used mounting medium of C. elegans embryos. Importantly, they demonstrate that the addition of iodixanol at the RIMM concentration is non-toxic to embryos. Using fluorescent C. elegans lines and confocal microscopy, the authors showed that imaging C. elegans embryos in RIMM can result in improved signal at deeper z-planes (further from the coverslip). This is presumably because the RIMM decreases spherical aberration or light scattering that occurs when sample and media have differing refractive indices. The authors subsequently demonstrated that images obtained with RIMM were more successfully segmented using 3D segmentation software than those acquired without RIMM. Finally, the authors used cellular morphometry from their images to show that the germ cell precursor P4 has increased sphericity, and therefore, likely increased cortical tension than its surrounding cell neighbors.

Major comments

How does tuning the RIMM under DIC conditions compare to other studies (Boothe et al.,) where phase contrast was used? I like that the authors used DIC imaging, because most C. elegans researchers image using DIC optics. However, the basis of matching the RIMM (according to Boothe et al., 2017) is minimizing the contrast in the phase images, and in DIC microscopy “Contrast can be varied instrumentally to suit the object” (Allen, David, and Nomarski, 1969). Does this present challenge in applying this RIMM across different microscopes with DIC or even on the same scope with the DIC adjusted differently? Due to results in other figures, it appears that it worked to tune the RIMM this way. Can the authors comment on the use of DIC optics vs. phase contrast for tuning RIMM?

I do like the quantification of loss of contrast in the iodixanol series, as opposed to the highly estimated method used in the other papers.

Minor comments

Methods:

In the Methods section, the authors mention that the embryos float in the RIMM (iodixanol) solution. Anyone who has previously worked with C. elegans embryos (which typically sink when mounting) would really want to know this. Thus, maybe mention of this could be made in the main text.

Explain in Methods that Ostu’s method is thresholding (this is explained in the figure, but not in the Methods for some reason).

The description of 3D segmentation using FIJI in the methods is very thorough. However, the 2D cell shape simulation description is incredibly minimal. Were there specific parameters or settings the authors used?

For the statistical analyses, was any software used?

Results:

When reporting “(0% iodixanol: 98%, n = 154; 30% iodixanol: 97%, n = 148)” specify that percentages refer to survival (I assume).

Figure 1:

I understand the desire to have zoomed in views of the embryos, but ideally, there would also be images with many embryos visible in the different iodixanol percentages, not just a single embryo (or two) shown per condition.

The viability of adult worms was not established. Can they tolerate the RIMM? This is relevant for longer-term imaging of adult C. elegans.

Figure 2:

I really liked that the authors quantified how well automated segmentation worked in the different conditions using ground truth cell counts.

Discussion:

“It should be noted that by coupling a RIMM and state-of-the-art 3D cell segmentation algorithms, such as RACE [24], 3DMMS [25], and BCOMS [26], researchers may obtain even better segmentation results than those shown in this study.” This is confusing – why didn’t the authors use these segmentation programs? What would “better” segmentation results look like?

The last sentence really fizzles out - maybe the authors could strongly restate the important findings of the paper.

Reviewer #3: In the manuscript by Xiong and sugioka, the authors systematically determine a refractive index (RI) matching solution for the imaging of C. elegans. This work builds from a previous study that introduced the used of iodixanol (OptiPrep) as a non-toxic method to increase the RI of various medias (Booth et al, eLife) and thus reduce spherical aberrations caused by RI mismatch that impedes fluorescent based imaging methods. Here the authors find that a 30% final solution of iodixanol enhances fluorescence imaging in C. elegans.

Overall this is a well presented work that is useful for many using C. elegans as a model system. I have only a few issues that should be addressed prior to publication.

Major suggestions:

1) The calculated Ri for the final solution is affected by the RI of the egg buffer (fixed at 1.33). Is this the actual RI of egg buffer? How critical is this?

2) On page 9, it is stated that “image quality is accessed. How is this done precisely? It seems to me that the absolute Histone signal (not only signal to noise) should be the same form each nucleus. How much loss is there in control vs. optimized?

3) The work here was completed using oil immersion. Do the authors expect a difference when using non oil (air, water, silicon, etc) optics? It would be helpful to make a comment on this.

Minor suggestions:

1) There are figure legends embedded in the main text, please remove.

2) In figure 2C is would be helpful (more clear) if the images for converted to gra

6. PLOS authors have the option to publish the peer review history of their article (what does this mean?). If published, this will include your full peer review and any attached files.

Reviewer #1: No

Reviewer #2: No

Reviewer #3: **Yes: **Paul Maddox

---

## [Author Response · Author response to Decision Letter 0]

23 Aug 2020

Dear editor and reviewers,

We appreciate your careful reading of our manuscript and insightful comments that greatly improved the overall quality of the revised manuscript. We also would like to thank the editor for giving us enough time to address these issues. We are delighted to inform you that we were able to obtain more data since the research curtailment of our institute has been partially lifted, and our new spinning disk confocal microscope with a silicone-immersion objective lens has been installed. We performed all key experiments that can answer the points raised by the reviewers, as summarized below.

1. We have performed more rigorous and precise analyses of RIMM-dependent improvement of signal intensity by imaging cell-cell boundary at different depths, at different iodixanol concentrations, with different types of objectives lenses.

According to the reviewers’ comments, we realized that the original Figure 2B-D did not successfully present the effect of RIMM. Thus, we have performed the precise depth-dependent analysis of signal intensity, as shown in the revised Figure 2C, 2D, 2E, and Figure S4. Furthermore, we confirmed that the RIMM-dependent improvement of signal intensity can be achieved using a silicone-immersion objective lens (new Figure 2D). Thus, RIMM can improve signal intensity at the deep cell layers regardless of the types of objective lenses.

2. We have analyzed whether molecules with the size of iodixanol can pass the eggshell.

While it has been known that C. elegans eggshell can pass small molecules with less than 1300 Da molecular weight, whether iodixanol (1550 Da) can pass the eggshell was not shown before. Thus, we have assessed and found that the fluorescent molecules with 1590 Da molecular weight penetrated into the eggshell as shown in Figure S2.

3. We have added more description of 2D cell shape simulation in Materials and Methods.

From here, we will answer each point raised by the reviewers (our answers are in blue).

Reviewer #1: The manuscript by Xiong & Sugioka describes the use of a refractive index matching medium to increase the visibility of in vivo fluorescence images of C. elegans embryos in order to facilitate the analysis of cellular morphometry. Although there are some potentially interesting aspects to this study, I consider it fundamentally flawed and not worthy of publication in its current form.

The eggshell of a C. elegans embryo is very impermeable and, as such, it is very unlikely that any of the iodixanol refractive index matching agent that was used will penetrate the embryo. This means that the only effect that refractive index matching the external medium will have is to reduce scattered light from the bathing medium/eggshell boundary. The claimed correction of spherical aberration in deep sections cannot occur. 

The eggshell of C. elegans is permeable to small molecules while the thin inner membrane attached to embryos, called permeability barrier, is not. A previous study by Olson et al., used three different sizes of fluorescent dyes to test their eggshell permeability. Olson et al. found that 389 Da and 1300 Da molecules passed the eggshell while that of 3000 Da did not (Olson et al., JCB 2012, PMID: 22908315). These small molecules penetrate into the perivitelline space— a space between eggshell and permeability barrier. The molecular weight of iodixanol is 1550 Da, only slightly higher than the reported limit of 1300 Da, such that it is likely that iodixanol can pass the eggshell. 

To test if molecules with the size of iodixanol can indeed pass the eggshell, we bathed embryos into 30% iodixanol containing Alexa 568-phalloidin (1590 Da). As shown in the new Figure S3, Alexa 568-phalloidin passed the eggshell. While it did not pass the permeability barrier, most areas of the embryo were in very close proximity to the surrounding perivitelline space. Iodixanol works without altering the intracellular refractive index (Boothe et al., eLife 2017). Rather, reducing the refractive index mismatch between embryos and the surrounding medium is sufficient to limit the depth-induced spherical aberration (Boothe et al., eLife 2017).

Based on the new data, we have added the text as follows:

Page 9, line 286:

“We assessed whether iodixanol could pass the C. elegans eggshell. C. elegans embryo is surrounded by the chitinous eggshell and the permeability barrier [22]. To reduce the RI mismatch between embryos and the surrounding media, 30% iodixanol may need to penetrate into the perivitelline space, a space between the eggshell and the permeability barrier (Figure S2; a space between a white solid line and a yellow dotted line). A previous study reported that 389 Da and 1300 Da fluorescent dye passed the eggshell while that of 3000 Da did not [22]. The molecular weight of iodixanol is 1550 Da, which is only slightly higher than the reported limit of 1300 Da, such that iodixanol likely passes the eggshell. To directly test if molecules with the size of iodixanol can pass the eggshell, we bathed embryos into 30% iodixanol containing Alexa 568-phalloidin (1590 Da). We found that Alexa 568-phalloidin passed the eggshell (Figure S2). Although the fluorescent molecules did not pass the permeability barrier, most areas of the embryo were in very close proximity to the surrounding perivitelline space (Figure S2). ”

In Fig. 2 the authors compare two embryos with different refractive index bathing media. There is very little difference to be seen, and both images exhibit the tell-tale deep section axial expansion typical of spherical aberration. There is also little visible improvement of deep section images to be seen in Fig. 3. Given the intrinsic variability of in vivo fluorescence between embryos and the fact that different embryos were used in the comparisons, I am unconvinced by these data. 

In the revised Figure 2, we performed more rigorous and quantitative analyses to prove that the iodixanol solution improved the signal intensity in the deep cell layers. According to our analysis, the membrane signal intensity was improved about 1.3-fold at 25 µm depth. Human eyes are known to outperform computer vision at recognizing patterns as discussed in the “Computer Vision in Cell Biology” authored by Gaudenz Danuser (Danuser, Cell 2011 PMID ). The following is a quote from this paper.

“Everyone who has programmed computer vision algorithms knows how hard it is to make a computer “see” the things our eye appears to recognize effortlessly. The reason for the ease with which the human vision system identifies relevant events in complex scenes is that our brain continually associates the observable image signals with our memory of previous visual experiences and our best interpretation of the scene.”

Computer vision is essential for stream-lined 3D cellular morphometry. The main aim of this manuscript is to show that iodixanol can let the computer vision, not the human vision, to better detect membrane signals.

There is a well-established way of reducing the spherical aberration from in vivo specimens that the authors do not mention, and that is to use water immersion lenses, where there is a reduced refractive index mismatch between cytosol and the lens immersion medium.

In our original manuscript, we have mentioned the silicone-immersion objective lens as a relatively costly measure to reduce the RI mismatch between samples and the lens immersion medium in the Discussion. Our view is similar for the water-immersion objective lens in terms of cost. To our knowledge, costs of a typical water-immersion objective (60x magnification, 1.2NA) and a 60x 1.3NA silicone immersion objective are around 10,000 USD and 12,000 USD, respectively. Theoretically, 1.2NA lens is 1.3-fold dimmer than the 1.3NA objective lens. Thus, we think that the silicone immersion objective lens is the better option for C. elegans research. We agree that we did not elaborate enough about this established method, and we modified the texts in the Introduction and the Discussion as shown below.

Page 3, line 46:

“Spherical aberrations misalign the optical paths, resulting in the loss of signal intensity and resolution. This effect increases with sample thickness, limiting the capacity of 3D cellular morphometry [1]. A major source of spherical aberration is the refractive index (RI) mismatch among three different materials: the lens immersion medium, samples, and the medium surrounding the samples. In typical C. elegans live imaging, samples are mounted onto 2%-4% agarose and are bathed in M9 or egg salt buffer [2][3], or bathed in these buffers without agarose [4]. These commonly used aqueous media (RI: ~1.33) should create the RI mismatch with both lens immersion media (RI: 1.52 for oil immersion) and C. elegans embryos (RI: 1.33-1.37; [5]).

In this study, we focused on the cost-effective method to reduce the RI mismatch that can be immediately adopted by the broad C. elegans research community. It is known that the use of water (RI: 1.33) and silicone (RI: 1.4) as lens-immersion media reduces the RI mismatch, but high-numerical aperture (NA) objective lenses designed for water- and silicone-immersion typically cost more than 10,000 USD.”

Page 18, line 554:

“The use of a RIMM in conjunction with a water- or a silicone- immersion objective lens would generate the best 3D segmentation results for confocal microscopy. This is because these objective lenses minimize the refractive index mismatch between 1) the objective lens immersion medium and samples and 2) the objective lens immersion medium and the medium surrounding samples. We have not compared the performance between water-immersion and silicone-immersion objectives, but the silicone-immersion objective would be better because of its higher numerical aperture and the very low evaporation rate of silicone.”

In addition, by using a silicone-immersion objective lens, we showed that a RIMM improved the depth-induced spherical aberration even when the RI mismatch between sample and the lens immersion medium was effectively minimal (revised Figures 2D and 2E). The result is consistent with the study by Boothe et al., which used both silicone-immersion objective lens and RIMM (Boothe et al., eLife 2017). These results together suggest that the benefit of RIMM is independent of the types of objective immersion media, and RIMM is a cost-effective solution to improve the spherical aberration.

I did find the author’s use of DIC microscopy to tune the refractive index of the medium interesting. As they demonstrate, this reduces the contrast extremes at the boundary of the embryo while maintaining the DIC contrast of structures within the embryo. This strategy might improve visibility of cells adjacent to the lateral sides of the embryo and thereby facilitate cell lineage studies. This simple technique might be worth developing into a technical note.

Thank you for your comment. 

Reviewer #2: Summary

The authors of this study apply refractive index media matching (RIMM), previously reported in Boothe, et al. 2017 for other organisms, to imaging of C. elegans embryos. They empirically and quantitatively determined the RIMM by titrating different percentages of iodixanol into egg salts, a commonly used mounting medium of C. elegans embryos. Importantly, they demonstrate that the addition of iodixanol at the RIMM concentration is non-toxic to embryos. Using fluorescent C. elegans lines and confocal microscopy, the authors showed that imaging C. elegans embryos in RIMM can result in improved signal at deeper z-planes (further from the coverslip). This is presumably because the RIMM decreases spherical aberration or light scattering that occurs when sample and media have differing refractive indices. The authors subsequently demonstrated that images obtained with RIMM were more successfully segmented using 3D segmentation software than those acquired without RIMM. Finally, the authors used cellular morphometry from their images to show that the germ cell precursor P4 has increased sphericity, and therefore, likely increased cortical tension than its surrounding cell neighbors.

Major comments

How does tuning the RIMM under DIC conditions compare to other studies (Boothe et al.,) where phase contrast was used? I like that the authors used DIC imaging, because most C. elegans researchers image using DIC optics. However, the basis of matching the RIMM (according to Boothe et al., 2017) is minimizing the contrast in the phase images, and in DIC microscopy “Contrast can be varied instrumentally to suit the object” (Allen, David, and Nomarski, 1969). Does this present challenge in applying this RIMM across different microscopes with DIC or even on the same scope with the DIC adjusted differently? Due to results in other figures, it appears that it worked to tune the RIMM this way. Can the authors comment on the use of DIC optics vs. phase contrast for tuning RIMM?

Phase-contrast microscopy converts differences in refractive index within the local area into image contrast. On the other hand, DIC converts differences in refractive index and sample thickness into image contrast. Boothe et al., made the right choice by selecting phase contrast for simplicity of analysis. We agree with the reviewer 2 that it is important to use DIC for C. elegans research community. As far as using samples with a similar thickness (e.g., C. elegans embryos), DIC image contrast should reflect the differences of refractive index. As the reviewer 2 pointed out, actual image contrast can be varied in different DIC settings, optical systems, and light intensities. Nevertheless, one should be able to reproduce the trend of our graph shown in Figure 1C to identify the iodixanol concentration which gives the lowest image contrast based on the principles described above. We elaborated this point in the text as follows:

Page 8, line 250:

“To determine the composition of a RIMM for C. elegans embryos, we imaged embryos bathed in egg salt buffer containing 0%, 10%, 20%, 25%, 30%, 40%, 50%, and 60% iodixanol (Figure 1 and Figure S1). A previous study used phase-contrast microscopy to determine the optimal iodixanol concentration, because it converts the local differences in refractive indices of transparent specimens into the image contrast [8]. The loss of image contrast indicates the RI matching between samples and the surrounding medium. In this study, we used Nomarski DIC microscopy as it is widely used by C. elegans researchers. DIC microscopy converts the local differences in thickness and refractive indices of transparent specimens into the image contrast [20]. C. elegans embryos are similar in thickness (30 µm; [21]), such that the changes in DIC image contrast should reflect the degree of RI mismatch between embryos and the surrounding medium. Thus, we sought a condition where the image contrast, calculated by the ratio of maximum to minimum signal intensity, were at their lowest (Figure 1A and 1C).”

We could not perform an experiment using a different DIC microscope because it is difficult to quantitatively say how two DIC settings are different. Instead, we have measured the signal intensity of TagRFP-PH at different iodixanol concentrations, as shown in Figure 2E. The graphs in Figure 1C and 2E show clear negative correlation. As fluorescent images were obtained without DIC optics, retrospectively one can assume that the graph in Figure 1C was effective in reporting the refractive index mismatch between samples and the surrounding media.

I do like the quantification of loss of contrast in the iodixanol series, as opposed to the highly estimated method used in the other papers.

Thank you for your comment. In the revised manuscript, we have directly measured the RI of these solutions and commonly used aqueous buffers using a refractometer to further make this study as a useful resource for the C. elegans research community (Figure S3). We have modified the text as follows:

Page 10, line 305:

“The RI of the 30% iodixanol solution was 1.379 ± 0.00004 at 22.5˚C according to the measurement using a refractometer at 589 nm wavelength (Figure S3). Selection of buffers should not have a large impact, since the RI of commonly used C. elegans buffers, such as M9 (RI: 1.335 ± 0.0001), egg salt (RI: 1.334 ± 0.00009), and Shelton’s growth medium (RI: 1.339 ± 0.00005) were similar at 22.5˚C (Figure S3). Based on these results, we concluded that 30% iodixanol, diluted in any common C. elegans buffers, is a RIMM for both C. elegans embryos and adult head regions.”

Minor comments

Methods:

In the Methods section, the authors mention that the embryos float in the RIMM (iodixanol) solution. Anyone who has previously worked with C. elegans embryos (which typically sink when mounting) would really want to know this. Thus, maybe mention of this could be made in the main text.

In the revised main text, the description can be found on page 11, line 338.

“Since embryos float in the 30% iodixanol solution, the distance between the objective lens and the embryos can differ when using 0% and 30% iodixanol. To minimize these differences in spherical aberration, we used polystyrene beads with embryo thickness as spacers (Figure 2A).”

Explain in Methods that Ostu’s method is thresholding (this is explained in the figure, but not in the Methods for some reason).

We have modified the description in the Materials and Methods as follows:

Page 6, line 191:

“For the generation of the binary images shown in Figure 2C, we used Otsu’s thresholding method with Fiji.”

The description of 3D segmentation using FIJI in the methods is very thorough. However, the 2D cell shape simulation description is incredibly minimal. Were there specific parameters or settings the authors used?

We have added the parameters used in the 2D simulation in the Materials and Methods as follows:

Page 7, line 227:

“First, we defined the position and size of thirteen cells by specifying the vertices and junctions in a two-dimensional space. Next, posteriorly located three cells were specified as “E.p”, “P4”, and “D” in a pattern resembles that of wild-type embryos. Other cells were specified as the same cell type (“Others”). Cells’ cortical tension (“outer boundary tension”) was set to 0.1 for all the cell types except for the P4 cell. Tension at the cell-cell boundary (“inner boundary tension”) was set to 0.1 for all cell-boundaries. In order to test the effects of P4 cortical tension on the P4 internalization, we used the P4 cortical tension 0.1, 0.15, and 0.2.”

For the statistical analyses, was any software used?

Thank you for pointing this out. We have added information about the software as follows:

Page 8, line 243:

“Statistical tests were performed using Prism 8 (Graph Pad).”

Results:

When reporting “(0% iodixanol: 98%, n = 154; 30% iodixanol: 97%, n = 148)” specify that percentages refer to survival (I assume).

We have modified the description as follows (changes underlined):

“(0% iodixanol: 98% hatched, n = 154; 30% iodixanol: 97% hatched, n = 148)”

Figure 1:

I understand the desire to have zoomed in views of the embryos, but ideally, there would also be images with many embryos visible in the different iodixanol percentages, not just a single embryo (or two) shown per condition.

We have added more embryo images in the revised Figure S1.

The viability of adult worms was not established. Can they tolerate the RIMM? This is relevant for longer-term imaging of adult C. elegans.

We have cultured L4 larva in 30% iodixanol solution for more than 24 hours. They normally produced hatching embryos. We have added the following description.

Page 10, line 303:

“L4-stage larva cultured in 30% iodixanol solution with OP50 E. coli produced fertile progenies, suggesting that the RIMM was not particularly detrimental to C. elegans postembryonic development.”

Figure 2:

I really liked that the authors quantified how well automated segmentation worked in the different conditions using ground truth cell counts.

Thank you for your comment.

Discussion:

“It should be noted that by coupling a RIMM and state-of-the-art 3D cell segmentation algorithms, such as RACE [24], 3DMMS [25], and BCOMS [26], researchers may obtain even better segmentation results than those shown in this study.” This is confusing – why didn’t the authors use these segmentation programs? What would “better” segmentation results look like?

The last sentence really fizzles out - maybe the authors could strongly restate the important findings of the paper.

These algorithms perform post-acquisition image processing such that they do not improve image quality at image-acquisition step. Thus, we consider that impact of our manuscript is independent of these methods. We have clarified this point as follows:

Page 18, line 565:

“With this low-cost solution, university instructors can also print 3D models and use them for biology curricula. It should be noted that a RIMM improves signal intensity at the image acquisition step, such that it can be coupled with other state-of-the-art post-acquisition 3D cell segmentation algorithms, such as RACE [26], 3DMMS [27], and BCOMS [28].”

Reviewer #3: In the manuscript by Xiong and sugioka, the authors systematically determine a refractive index (RI) matching solution for the imaging of C. elegans. This work builds from a previous study that introduced the used of iodixanol (OptiPrep) as a non-toxic method to increase the RI of various medias (Booth et al, eLife) and thus reduce spherical aberrations caused by RI mismatch that impedes fluorescent based imaging methods. Here the authors find that a 30% final solution of iodixanol enhances fluorescence imaging in C. elegans.

Overall this is a well presented work that is useful for many using C. elegans as a model system. I have only a few issues that should be addressed prior to publication.

Major suggestions:

1) The calculated Ri for the final solution is affected by the RI of the egg buffer (fixed at 1.33). Is this the actual RI of egg buffer? How critical is this?

The buffer we have used is the “egg salt buffer” which contains 118 mM NaCl and 48 mM KCl. We have used this buffer for imaging since it is more physiologically isotonic to embryos than M9 buffer (Edgar LG, Methods Cell Biol. 1995; PMID: 22226523). 

The effects of salts used in typical physiological buffers were supposed to be very minimal according to the previous study that estimated the RI of NaCl solution (Kamal and Esmail, Optical Materials 1993, pp.195-199.). Nevertheless, we have directly measured the RI of iodixanol solutions and commonly used C. elegans buffers, using a refractometer, since estimation of accurate RI of complex solution was not trivial (Figure S3). We have added relevant information as follows:

Page 10, line 305:

“The RI of the 30% iodixanol solution was 1.379 ± 0.00004 at 22.5˚C according to the measurement using a refractometer at 589 nm wavelength (Figure S3). Selection of buffers should not have a large impact, since the RI of commonly used C. elegans buffers, such as M9 (RI: 1.335 ± 0.0001), egg salt (RI: 1.334 ± 0.00009), and Shelton’s growth medium (RI: 1.339 ± 0.00005) were similar at 22.5˚C (Figure S3). Based on these results, we concluded that 30% iodixanol, diluted in any common C. elegans buffers, is a RIMM for both C. elegans embryos and adult head regions.”

2) On page 9, it is stated that “image quality is accessed. How is this done precisely? It seems to me that the absolute Histone signal (not only signal to noise) should be the same form each nucleus. How much loss is there in control vs. optimized?

In the original analysis, we have quantified the ratio between peak histone-gfp signal intensity and the mean background signal intensity. However, these signals were measured in a range of 15 µm and 30 µm depths. Since nuclear position and sizes are different among images used for measurement, we realized that the original analysis might not be precise, as the reviewer 2 pointed out. Thus, in the revised Figure 2, we measured the TagRFP-PH membrane signal at 5-µm, 15-µm, and 25-µm depth. We have provided more detailed explanation in Figure S3 as to how this measurement was performed. 

3) The work here was completed using oil immersion. Do the authors expect a difference when using non oil (air, water, silicon, etc) optics? It would be helpful to make a comment on this.

We were interested in performing the suggested comparison before our original submission, but were not able to do due to COVID-19. Recently our microscope equipped with a silicone-immersion objective lens has been installed, and we performed the same analysis we have done with the oil-immersion objective. As shown in the revised Figure 2D, we observed the similar extent of improvement in terms of signal intensity. Please note that the configuration of microscopes used for the original analysis (a Leica microscope equipped with an oil-immersion objective and EMCCD cameras) and the revised analysis (an Olympus microscope equipped with a silicone-immersion objective and a sCMOS camera) were slightly different. Thus, we cannot directly compare the graphs in the revised Figure 2D.

Minor suggestions:

1) There are figure legends embedded in the main text, please remove.

According to our understanding, Plos One’s author instruction requires us to insert these figure legends in the first-referred position. We will correct this if it were wrong.

2) In figure 2C is would be helpful (more clear) if the images for converted to gra

We agree that the image may be better in grayscale. In the revised Figure 2C, we have provided more clear evidence of RIMM’s role in reducing the depth-induced spherical aberration. And we have replaced the images in Figure 3B and 3C as grayscale to better show the differences in signal and noise.

---

## [Editor Report · Decision Letter 1]

27 Aug 2020

Improved 3D cellular morphometry of Caenorhabditis elegans embryos using a refractive index matching medium

PONE-D-20-18786R1

Dear Dr. Sugioka,

We’re pleased to inform you that your manuscript has been judged scientifically suitable for publication and will be formally accepted for publication once it meets all outstanding technical requirements. Thank you for sending this interesting story to PLOS ONE.

Within one week, you’ll receive an e-mail detailing the required amendments. When these have been addressed, you’ll receive a formal acceptance letter and your manuscript will be scheduled for publication. In addition to any required amendments you receive, please be sure to subscript the numbers 1 and 4 in "P1" and "P4" in the final manuscript, because non-subscripted P1 and P4 are postembryonic P blast cells in C. elegans.

Kind regards,

Bob Goldstein

Academic Editor

PLOS ONE

---

## [Editor Report · Acceptance letter]

16 Sep 2020

PONE-D-20-18786R1 

Improved 3D cellular morphometry of *Caenorhabditis elegans* embryos using a refractive index matching medium 

Dear Dr. Sugioka:

I'm pleased to inform you that your manuscript has been deemed suitable for publication in PLOS ONE. Congratulations! Your manuscript is now with our production department. 

Kind regards, 

on behalf of

Dr. Bob Goldstein 

Academic Editor

PLOS ONE